# Mass Spectrometry Contribution to Pediatric Cancers Research

**DOI:** 10.3390/medicina59030612

**Published:** 2023-03-20

**Authors:** Marco Agostini, Pietro Traldi, Mahmoud Hamdan

**Affiliations:** Istituto di Ricerca Pediatrica Città della Speranza, Corso Stati Uniti 4, 35100 Padova, Italy

**Keywords:** pediatric cancer, two-dimensional gel electrophoresis, MS-based proteomics, metabolomics, pharmacokinetics of anticancer drugs, epigenetics, chromatin modifications

## Abstract

For over four decades, mass spectrometry-based methods have provided a wealth of information relevant to various challenges in the field of cancers research. These challenges included identification and validation of novel biomarkers for various diseases, in particular for various forms of cancer. These biomarkers serve various objectives including monitoring patient response to the various forms of therapy, differentiating subgroups of the same type of cancer, and providing proteomic data to complement datasets generated by genomic, epigenetic, and transcriptomic methods. The same proteomic data can be used to provide prognostic information and could guide scientists and medics to new and innovative targeted therapies The past decade has seen a rapid emergence of epigenetics as a major contributor to carcinogenesis. This development has given a fresh momentum to MS-based proteomics, which demonstrated to be an unrivalled tool for the analyses of protein post-translational modifications associated with chromatin modifications. In particular, high-resolution mass spectrometry has been recently used for systematic quantification of chromatin modifications. Data generated by this approach are central in the search for new therapies for various forms of cancer and will help in attempts to decipher antitumor drug resistance. To appreciate the contribution of mass spectrometry-based proteomics to biomarkers discovery and to our understanding of mechanisms behind the initiation and progression of various forms of cancer, a number of recent investigations are discussed. These investigations also include results provided by two-dimensional gel electrophoresis combined with mass spectrometry.

## 1. Introduction

Despite impressive advances in the genomic, epigenetic, and proteomic fields, various forms of cancer continue to exact unacceptable price in terms of deaths and the quality of life throughout the world. On the other hand, the last three decades have witnessed intense research activities both at the clinical and at the academic levels to identify reliable biomarkers, more targeted and less toxic therapies, and diagnostic strategies which can help in the early diagnosis and better prognosis of these devastating diseases. One major line of investigation concentrated on the identification of proteins which under disease conditions may experience significant change in their levels of expression. These studies involved both adult and pediatric patients delivering much needed information, which may lead to new drug targets. It is worth pointing out that these proteomic studies assume more relevance in the case of pediatric patients, because pediatric cancers tend to have less genomic alterations compared to adults, while proteomic changes are comparable to those observed in adult cancers. These proteomic analyses can be performed either on tissue samples [1,2] or on biofluids [3,4]; the latter approach is commonly referred to as liquid biopsy. A wide range of liquid samples used in this approach can be obtained in a non-invasive manner, which renders it highly attractive in clinical investigations, particularly in pediatric analyses. The most commonly analyzed biofluids are blood (plasma, serum) and urine, but other biofluids have been examined by MS-based proteomics including expressed prostatic secretions [5], saliva [6], tears [7], and cerebrospinal fluid (CSF) [8]. In a recent review by McEachron and Helman [9], the authors underlined the emerging role of liquid biopsy as a versatile source for various biomarkers, which are necessary for monitoring and management of disease both in adults and in pediatric patients. According to the authors, the increasing use of this approach removed the obstacle of performing serial biopsies in pediatric patients. Such serial analyses allow monitoring and management of solid and CNS tumors by using minimally invasive tests. These analyses can detect both cellular and molecular biomarkers of a given disease, including cell-free DNA, circulating tumor cells, micro RNAs, proteins, and metabolites.

Currently, the identification of the observed proteins may be established by monitoring DNA and mRNA sequences. However, such approach carries two relevant limitations. First, it cannot provide accurate information on the levels of expression of the proteins in question [10,11], and second, it does not furnish information on their stability or on any possible post-translational modifications [12,13]. Such limitations underlined the need for an alternative, yet complementary approach to protein assessment based on transcripts. MS-based proteomic techniques provide such second approach in which various proteomic strategies can be used to assess protein expression, assign function(s), and identify possible post-translational modifications, which a given protein may experience. These strategies employ a wide range of MS-based methodologies and a variety of separation and sample preparation protocols. It is relevant to point out that such strategies are not exclusively designed to identify proteins as biomarkers. Other biological molecules present in blood, cerebrospinal fluid, and tissues can have an equally important role as possible biomarkers of various forms of cancer. In the last thirty years, a battery of separation methods coupled to high-resolution mass spectrometry detection have been used to explore the molecular mechanisms responsible for the initiation and progression of various forms of cancer. The capability of some of these separation methods has been enhanced through the additional use of purification, fractionation, metabolic labelling, and dominant proteins depletion to favor the detection of low-copy proteins.

The recent review by McEachron and Helman [9] described the present status of key aspects of pediatric cancer research, focusing on genetic and epigenetic drivers of disease, cellular origins of various pediatric cancers, and cellular immunotherapies. The conclusions by these authors are highly informative and deserve to be underlined in the present work: First, at the genetic level, the authors argued that many genetic changes identified in pediatric cancers are distinct from adult cancers and will require unique interventions. Second, it is becoming clear that pediatric tumors appear to have a high degree of epigenetic changes that lead to a widespread alteration in gene expression secondary to these changes, understanding such alterations may lead to the discovery and development of novel therapies. Third, germline genetic variations associated with predisposition to cancer appear to occur at a higher frequency in pediatric patients compared to adult cancer patients.

Over the last two decades, a strong body of evidence has emerged confirming that the initiation and progression of cancer is controlled by both genetic and epigenetic events. Unlike genetic alterations, which are almost impossible to reverse, epigenetic aberrations are potentially reversible, allowing the diseased cell population to return to its normal state. The X-ray crystal structure of chromatin [14] revealed a structure made of repeating units of nucleosomes, which consist of ~146 base pairs of DNA wrapped around an octamer of 4 core histone proteins (H3, H4, H2A, and H2B) (Figure 1). 

Genetic and epigenetic investigations that followed provided sufficient information to suggest that chromatin structure can be modified in a number of ways. The recent literature cites at least eight distinct types of modifications found on histones, where a major part of available information regards small covalent modifications: methylation, acetylation, and phosphorylation [15]. MS-based strategies are the methods of choice for the analyses and quantification of protein post-translational modifications (PTMs), including those associated with histones. Currently, “shot gun” MS-based strategy is the preferred choice for the analyses and identification of histones PTMs. This strategy encompasses three different methods: In bottom-up method, histones are enzymatically digested, and the resulting peptides mixture is subjected to reversed-phase liquid chromatography (LC) separation prior to MS and MS/MS analyses. The LC step is highly critical, in which various gradients have to be used to separate a high percentage of isobaric peptides present in the investigated digests [16]. The main limitation of this method is due to the presence of a high number of basic residues within the histone sequence. This limitation is often resolved by using Arg-C protease instead of trypsin [17]. Top-down or middle-down are two MS-based methods, which can be used for the analyses of intact histones or large protein fragments, respectively [18]. In top-down, intact histones are chromatographically separated and directly analyzed, while in middle-down, long histone peptides (>5 kDa) are obtained through enzymatic digestion with proteases that cleave at less frequently occurring residues, such as Glu-C and Asp-N, and are usually separated using weak-cation exchange combined with hydrophilic interaction liquid chromatography [19]. One of the major challenges for both methods is the ability to distinguish isobaric peptides, a problem that becomes more relevant for the larger molecules analyzed in middle- and top-down methods. Using these methods, isobaric species are often co-isolated and fragmented at the MS/MS level, and as a consequence, they cannot be discriminated. Another issue related to top- and middle-down methods is that long peptides have much wider charge state distributions than bottom-up peptides, which reduces the overall signal intensity of each charged state. This sensitivity issue may be partially mitigated for middle-down method by prefractionation of the digested mixture. In this article, an attempt is made to underline the contribution of MS-based strategies to cancers research. The authors cite and discuss both early as well as recent studies in which the use of mass spectrometry contributed to the discovery of potential cancer biomarkers, understanding of the progression, and the biology of various tumors. A particular emphasis is given to the role of MS-based studies of chromatin modifications, which are central to the understanding of various mechanisms responsible for different epigenetic alterations. 

## 2. MS-Based Proteomic Analyses

### 2.1. Two-Dimensional Gel Electrophoresis Combined with Mass Spectrometry

Currently, there are two main strategies for the analyses and identification of proteins (proteoforms) present in complex biological samples: liquid chromatography coupled to mass spectrometry (LC-MS, MS-MS) and two-dimensional polyacrylamide gel electrophoresis in combination with mass spectrometry (2D-PAGE-MS). The first approach is commonly called shotgun proteomics, which encompasses the methods of top-down, middle-down, and bottom-up [20]. Before discussing some data generated by the two strategies, the following observations are of interest: (i) It is not always appreciated that the two strategies are complementary (see Figure 2).

The intact proteins separated by 2D-PAGE are often digested into peptides to be separated and identified by LC/MS, MS-MS. (ii) Two-dimensional gel electrophoresis was instrumental in the birth of modern proteomics over 40 years ago, an observation which explains why some research scientists consider such approach as outdated technique for current proteomic analyses. The authors will not enter into the technical details, advantages, and drawbacks of both strategies. Instead, a well-written review on what two-dimensional gel electrophoresis can offer in present-day proteomics is worth reading [21]. 

Polyacrylamide gel electrophoresis (2D-PAGE) [22] remains one of the main players in the investigation of complex proteomic systems. A modified version of this technique is known as two-dimensional differential in gel electrophoresis (2D-DIGE) [23], where the proteins are pre-labelled with up to three fluorescent dyes. This labelling procedure allows different protein samples to be run on the same gel with the direct consequence of a better reproducibility and easier quantification when compared with 2D-PAGE. This technique combined with MALDI-MS was used by Braoudaki et al. [24] to investigate the proteomic profiles of pediatric patients of acute lymphoblastic leukemia (ALL). The scope of the study was the search for candidate biomarkers in low- and high-risk patients, which could be used for diagnosis, prognosis, and patient-targeted therapy. The authors used Western blot analyses to confirm the differential expression of a number of the detected proteins. Bone marrow and peripheral blood plasma and cell lysates samples were obtained from pediatric patients with low- and high-risk ALL. To favor the detection of low-abundant proteins, depletion of dominant proteins was performed. This study identified a number of proteins, which the authors described as potential biomarkers for the stratification of ALL. There are two interesting elements in this study, which should be underlined: First, proteins depletion in complex biological samples helps in the detection of low-copy proteins. Second, differentially expressed proteins, identified in the same study, were used for pathway analysis. For this purpose, the Swiss-Prot accession numbers were inserted into the search tool for the retrieval of interacting genes/proteins. Second, differentially expressed proteins detected in 2D-PAGE and in MALDI-MS analyses were further confirmed by Western blot analyses.

Cancer cell resistance to chemotherapy is still considered a major factor in the failure of treatment of different forms of pediatric cancer, in particular high-risk forms of the disease. A number of works have shown that such failure can have various forms. For example, some forms of cancer can be intrinsically resistant and unlikely to respond to such treatment, whereas other forms show initial response to such treatment but then regrow to become resistant. The question of chemoresistance remains central for the success or failure of a given treatment and more concentrated efforts are needed to understand the main factors/molecules which may trigger such resistance. 

In one of the early studies by Sinha et al. [25], 2D-PAGE in combination with MALDI-TOF-MS was used to investigate chemoresistance development in melanoma cell lines. The authors used a panel of human melanoma cell variants exhibiting low and high level of resistance to four commonly used anticancer drugs in melanoma treatment. The authors reported that a number of proteins have experienced differential expression. For example, in most cell variants, upregulation of heat shock protein isoforms HSP60 and HSP70 was observed. The same study reported an increased expression of the small stress protein HSP27. It is interesting to note that in an earlier study by Sarto et al. [26], it was reported that upon cell stimulation in leukemia, HSP27 was frequently the target of phosphorylation. The same protein has been frequently associated with the inhibition of apoptosis induced by different chemotherapeutics.

It is well established that cisplatin-based chemotherapy is one of the main treatments for various forms of tumors in advanced stage. Although the rate of initial response to the treatment is usually high, relapse with cisplatin-resistant cells occurs in the majority of patients. It is now well recognized that chemoresistance is a major problem in cancer treatment; therefore, proteomic analyses, which can shed some light on the role of different proteins involved in such resistance, are much needed. A number of studies have demonstrated that cisplatin-based agents are known to target DNA, with which can form both inter-strand and intra-strand crosslinks [27]. Further studies have also found that mitochondria were associated with the cisplatin resistance. Mitochondrial DNA and membrane proteins were reported as preferential targets of cisplatin. It was also shown by several groups that mitochondria impairment appeared to play an important role in the platinum resistance of ovarian cancer cells. Using 2D-DIGE combined with MALDI-TOF-MS, Dai et al. [28] investigated the mitochondrial proteins difference between platinum-sensitive human ovarian cancer cell with that of four platinum-resistant variants. The authors reported a number of proteins observed in the resistant cells, which have experienced up or downregulation. The same study reported that some proteins in the resistant cells were upregulated more than three folds compared to their counterparts in platinum-sensitive cells. It is also fair to say that regardless of its entity, cancer is a highly complex disease composed of cancer cells, stromal cells, and the extracellular matrix. It is now commonly accepted that the genetic and epigenetic alterations could be responsible for the initial growth and progression in various types of cancer. However, it is not clear yet whether the microenvironment (stromal cells, extracellular matrix) plays a role in the resistance to cell apoptosis. If that is the case, then it would not be unreasonable to assume that the microenvironment of a given cancer can influence its progression, its drug resistance, and metastasis. A recent review by Dzobo [29] has given an account on how deciphering the role(s) of the tumor microenvironment could lead to more effective cancer therapies. The author argued that therapies targeting exclusively cancer cells are usually insufficient when the stromal component of the microenvironment causes therapy resistance. In other words, the therapy is likely to be more successful if the anticancer drug can be designed to target both cancer as well as stromal cells.

2D-PAGE, DIGE in combination with mass spectrometry have furnished extensive experimental data on the role of proteomics in the understanding of various diseases, in particular various forms of cancer. Having said that, this powerful analytical tool like many others has its limitations, which are mainly due to extensive heterogeneity and a wide dynamic range of proteins present in a given proteomic system. Other limitations are associated with sample load in 2D gel analyses and the width of the pH range in which the separation takes place. In a well-designed study, Gygi et al. [30] evaluated the capability of 2D-PAGE in detecting low abundance proteins in yeast proteome. The authors used this technique to separate and visualize proteins followed by mass spectrometry for their identification. The main conclusion of this study was that despite the high sample load and extended electrophoretic separation, proteins from genes with codon bias values of <0.1 (low abundance proteins) were not found; this is despite existing data indicating that at least half of all yeast genes fall into that range. Low abundance proteins were however detected when higher sample loads were used. This included beginning with 50 mg of total yeast protein as well as using a strategy that included SDS/PAGE, in-gel digestion, strong cation exchange chromatography separation, and on-line microcapillary LC-MS/MS techniques.

The results of this study are in line with other investigations, which highlighted the complexity of proteins heterogeneity in terms of the range of their isoelectric points, molecular masses, and abundances, which can challenge the most powerful and high-resolution techniques including 2D-PAGE. It is sufficient to recall that in a standard human cell, the most abundant protein is often actin, which is present at roughly 10^8^ molecules per cell compared with some cellular receptors or transcription factors, which are probably present at 10^2^–10^3^ molecules per cell [31]. The situation can be more challenging in sera, where albumin is at 40 mg/mL, while cytokines are present at pg/mL levels. It has been known for a number of years that to detect low-copy proteins in a proteome using 2D-PAGE, enrichment or prefractionation strategies are needed. What these and other similar studies tell us is that the combination of 2DE-MS can separate and identify hundreds of proteins in a single analysis; however, to reach low copy proteins within the same analysis, other strategies have to be adapted, including fractionation and enrichment on the 2DE side, while the MS component has to adapt all recent developments not only on the MS side but also on the chromatography of the digested proteins separated by the 2DE.

Drug resistance remains one of the big challenges in oncology, particularly for cytotoxic chemotherapy. In the last few years, the use of 2D-DIGE to investigate drug resistance has diminished in favor of LC/MS, MS-MS and mass spectrometry imaging (MSI) using MALDI ionization. MSI is a label-free method that is used to map the distribution of a wide range of molecules in different biological samples. One of the key features of MALDI-MSI that makes its use appealing is the ability to detect and study the distribution of multiple compounds simultaneously without the need for labelling. It is of interest to note that the powerful and widely used LC/MS-MS lacks information on spatial distribution, which is highly important in the analyses of heterogeneous biological samples. Compared with traditional clinical imaging methods requiring the use of indirect labels such as radiolabels or probes, MSI not only has high spatial resolution in low micrometer scale but also can simultaneously detect the spatial distribution of a drug and its resulting metabolites; both features are relevant in studies related to drug mechanisms of action and related drug resistance [32]. Traditional imaging methods, including whole body autoradiography (WBA), positron emission tomography (PET), and fluorescence microscopy (FM), have all been used to study drug distribution in pre-clinical models and occasionally in patients. These methods have the advantage of being less invasive than biopsy-based assays, but both WBA and PET have low spatial resolution and require radiolabeled drugs. FM allows the study of drug distribution at a cellular level but is limited to drugs that have intrinsic fluorescence; otherwise, fluorescent labels have to be introduced. Such labelling procedure has its drawbacks, where the resulting images fail to distinguish between the parent drug and its metabolites. Like any other analytical technique, MSI has a number of critical steps, which have to be rigorously implemented before this relatively young technique can deliver on its potential. The choice of the matrix and its application to the sample are critical steps for the success of analyses. Interference of the matrix ions with those of the sample, particularly at low *m*/*z* values, is a well-known defect of MALDI. This defect is normally addressed by high-resolution MS to investigate the suspected ions; in the absence of such option, the use of different matrices may provide a partial solution. MSI generate impressive amounts of data, which have to be carefully analyzed and interpreted, a step requiring experience together with powerful informatic tools, including suitable algorithms.

There is an ever-growing body of evidence suggesting that epigenetic mechanisms play an important role in bringing about drug resistance in cancer cells. Furthermore, a number of recent studies have shown some correlation between antitumor drug resistance and some epigenetic alterations [33,34]. Pediatric acute lymphoblastic leukemia (ALL) is a representative example on how these recent data are going to impact our understanding of antitumor drug resistance. Survival rates for children diagnosed with ALL have improved substantially, but patients who relapse continue to face an unacceptable prognosis. Genomic research allowed for considerable advances in risk stratification of ALL and have contributed to the development of novel therapeutic strategies. These strategies have been enhanced and extended by the new discoveries in epigenetic research. Such discoveries indicate an important role for the epigenome as a mediator of disease relapse and chemoresistance. Furthermore, genome-wide approaches to interrogate the epigenetic landscape have revealed relapse-specific patterns of epigenetic aberrations that tightly correlate with gene expression. Evidently, further studies are needed to better understand the correlation between these epigenetic alterations and leukemogenesis and how these abnormalities contribute to differences in chemotherapy sensitivity. Many preclinical and early phase studies have concentrated on the reversal of the epigenetic alterations. Some of these studies have demonstrated the validity of such approach; however, the application of this approach at the clinical level only had a partial success, mainly because of undesired side effects [35]. Considering the fast progress in the area of novel epigenetic modulators, there is a high potential for near future discovery of more efficient agents necessary to apply this epigenetic therapeutic approach in ALL.

Gel electrophoresis in combination with mass spectrometry have furnished extensive experimental data on the role of proteomics in the understanding of various diseases, in particular various forms of cancer. Having said that, this powerful analytical tool like many others has its limitations, which are mainly due to extensive heterogeneity and a wide dynamic range of proteins present in a given proteomic system. Other limitations are associated with sample load and the width of the pH range in which the separation takes place. In a well-designed study, Gygi et al. [30] evaluated the capability of 2D-PAGE in detecting low abundance proteins in yeast proteome. The authors used this technique to separate and visualize proteins followed by mass spectrometry for their identification. The main conclusion of this study was that despite the high sample load and extended electrophoretic separation, proteins from genes with codon bias values of <0.1 (low abundance proteins) were not found; this is despite existing data indicating that at least half of all yeast genes fall into that range. Low abundance proteins were however detected when higher sample loads were used. This included beginning with 50 mg of total yeast protein as well as using a strategy that included SDS/PAGE, in-gel digestion, strong cation exchange chromatography separation, and on-line microcapillary LC-MS/MS techniques.

The results of this study are in line with other investigations, which highlighted the complexity of proteins heterogeneity in terms of the range of their isoelectric points, molecular masses, and abundances, which can challenge the most powerful and high-resolution techniques. It is sufficient to recall that in a standard human cell, the most abundant protein is often actin, which is present at roughly 10^8^ molecules per cell compared with some cellular receptors or transcription factors, which are probably present at 10^2^–10^3^ molecules per cell [31]. The situation can be more challenging in sera, where albumin is at 40 mg/mL, while cytokines are present at pg/mL levels. It has been known for a number of years that to detect low-copy proteins in a proteome using 2D-PAGE, enrichment or prefractionation strategies are needed. What these and other similar studies tell us is that the combination of 2D-PAGE-MS can separate and identify hundreds of proteins in a single analysis; however, to reach low copy proteins within the same analysis, other strategies have to be adapted, including fractionation and enrichment.

### 2.2. Mass Spectrometry Analyses of Biofluids

The invasive nature of surgical biopsies prevents their sequential application to monitor disease, particularly for pediatric patients. Currently, there is enough evidence to suggest that these single biopsies fail to reflect cancer progression, intratumor heterogeneity, and drug sensitivities, which tend to change over time. While tissue samples have the potential to provide novel biological insights, many clinical proteomic studies have chosen clinical samples that are obtainable in a non-invasive or minimally invasive manner, as in the case of liquid biopsies. In recent years, liquid biopsy has assumed an important role in disease monitoring both in adult and in pediatric patients. Furthermore, it is becoming more evident that such monitoring is central in the management of various forms of cancer, particularly in the pediatric population. The recent and unmistakable increase in the use of this powerful analytical tool is due to its minimally invasive tests that can be performed on easily accessible body fluids. Prior to the development of this tool, the repetition of surgical biopsies in pediatric patients was discouraged unless considered medically necessary. Such limitation hindered the search for reliable biomarkers associated with the development of certain diseases. Such lack of reliable biomarkers was more evident in studies related to CNS and solid tumors in children. The recent literature suggests that the use of liquid biopsies could greatly reduce sampling limitations and thus allowing time-dependent assessment of disease evolvement. Encouraging data generated by liquid biopsy sampling has been reported recently, where the detection and quantification of cfDNA have been revealed in various solid and CNS tumors of childhood [36].

Chromatography coupled to high-resolution mass spectrometry is one of the principal tools for the analyses of samples derived from biofluids. Because of the very complex nature of many biological samples, efficient sample preparation protocols to remove unwanted components and to selectively extract the compounds of interest are an essential part of almost every bioanalytical workflow. The proteomes of biofluids, including serum, saliva, cerebrospinal fluid, and urine, are rich with proteins and other biomolecules varying depending on the physiological and/or pathophysiological of the subject. Advances in mass spectrometric technologies have facilitated the in-depth characterization of biofluid proteomes which are now considered hosts of a wide array of clinically relevant biomarkers for a number of diseases [37]. Currently, there are two main strategies in the proteomic analyses of biofluids [38]. The strategy for biomarkers discovery requires multidimensional fractionation in combination with high-resolution mass spectrometry methods, including tandem mass spectrometry (MS/MS). While this approach provides a large amount of data and can identify hundreds of proteins, it remains time consuming and limited in the number of comparisons that can be made. On the other hand, the high throughput nature of the diagnostic development approach (see Figure 3) has made it highly popular.

In this method, biofluids from hundreds of patients and controls are analyzed by LC/MS, MS/MS. Information on the detected proteins can be rapidly obtained, while the use of sophisticated algorithms facilitates the recognition of various peaks within a multitude of spectra that allow the source of the biofluids (i.e., disease-afflicted or healthy individual) to be established. Although this approach has shown remarkable ability to correctly diagnose patients in blind validation studies, it lacks the capability to directly identify specific proteins associated with the peaks that indicate the source of the biofluids. In cases where the diagnostic protein peaks have been identified, their relationship to the disease condition is not clear because the proteins within the peak are not identified. 

It can be said that one of the main objectives of various MS-based proteomic analyses of biofluids is to detect and possibly identify molecular entities associated with a given disease condition. These (entities can be protein(s), metabolite(s), lipids, cfDNA, or other biomolecules present in the investigated sample. Some of these molecules are further investigated to establish their potential as valid biomarkers for a given disease. This phase of discovery is commonly followed by a phase of validation and clinical testing of these potential biomarkers. It has to be emphasized that a successful conclusion of both phases requires highly strict standard operating procedures. The application of such procedures becomes fundamental in metabolic analyses, where a wide spectrum of methods is used in different metabolomics laboratories. In view of this, the Metabolomics Society has set up the “metabolomics Standards Initiative Committee that has established rules to standardize metabolomics systems [39]. Quality control and standard operating procedures should be carefully followed to reduce pre-analyses errors.

Research efforts over the last two decades have provided strong evidence to suggest that various forms of cancer can be attributed to metabolic alterations in various cellular systems. These emerging data have been supported by recent research efforts in which new technologies including high-resolution mass spectrometry have been employed. A representative example of such efforts has been recently reported by Petrik et al. [40]. In their study, the authors used high-resolution negative-ion electrospray mass spectrometry to analyze blood spots taken from newborns. Untargeted metabolomic analyses were performed on 48 archived blood spots from newborns who later developed AML as well as samples taken from 46 healthy controls. Before considering the main conclusions of this study, the following considerations should be underlined: First, ongoing research activities that consider cancer as a disease of highly altered cellular metabolism have accelerated interest in snapshot metabolomics in various human tissues. Second, the chemistry of the endogenous metabolites is fairly complex, which ranges from charged inorganic molecules to hydrophobic lipids to hydrophilic carbohydrates. Because of such complexity, no single analytical method can capture the full range of endogenous metabolites. This chemical complexity required the combination of two powerful analytical tools: high-resolution mass spectrometry and liquid chromatography. The chromatography component allows the separation of various chemical entities, which happen to have different chemical properties (different hydrophobicity/hydrophilicity), while high-resolution mass spectrometry provides accurate molecular weight assignments of the separated metabolites. The authors concluded that metabolic profiles of newborns’ dried blood spots could discriminate between newborns that later developed acute myeloid leukemia (AML) from those that did not, and that these profiles were sex specific. Furthermore, predictive metabolites in females suggested that several biological pathways may be involved in AML initiation. One pathway contained four ceramide metabolites positively associated with AML risk, while another pathway in females may be linked with protective effects of breastfeeding. Metabolite predictors in males were more heterogeneous and showed no relationship with the available covariate risk factors, suggesting that etiology in males may be more multifactorial than females. The authors also acknowledged that the sample size was limited in accordance with the rarity of pediatric AML. Furthermore, annotations of the metabolite predictors were limited by the absence of tandem mass spectrometry data for confirmation. 

In a more recent study [41], LC/MS was used to investigate metabolites in urine samples taken from medulloblastoma (MB) patients. Samples were taken from over 100 (MB) patients and from a similar number of healthy controls. These analyses concluded that among all differentially identified metabolites, four metabolites (tetrahydrocortisone, cortolone, urothion, and 20-oxo-leukotriene E4) were specific to MB. Furthermore, the analysis of these four metabolites in pre- and postoperative MB urine samples showed that their levels returned to a healthy state after the operation (especially after one month), showing the potential specificity of these metabolites for MB.

## 3. Investigation of Pediatric Leukemia by Mass Spectrometry-Based Methods

### 3.1. Acute Myeloid Leukemia

Acute lymphocytic (lymphoblastic) leukemia (ALL) is the more diffused form of leukemia, where three out of four childhood cases are attributed to this form, in which the predominant phenotype is B-cell precursor ALL. It affects children in the age from 2 to 5 years, while the less common T-cell phenotype tends to increase with age (United States SEER program 1975–1995 Vol. vi). B-cell precursor ALL tends to have numerical and structural chromosomal abnormalities such as high hyper diploidy (defined by the presence of 51–67 chromosomes, referred to as 51+). Acute myeloid leukemia (AML) is rare in children and occurs uniformly across all ages. This form of leukemia is often characterized by recurrent chromosomal abnormalities such as MLL fusions located at chromosome 11q23 [42]. It is useful to emphasize that world-wide clinical investigations have demonstrated that ALL is an intrinsically lethal form of cancer. Having said that, it is encouraging to note that current cure rates for ALL using combination chemotherapy are around 90%, making such achievement one of the success stories of oncology. 

Almost 50 years ago, French-American-British (FAB) Co-operative Group published proposals for the classification of the acute leukemias [43]. These proposals based on conventional morphological and cytochemical methods were suggested following the study of peripheral blood and bone marrow films from some 200 cases of acute leukemia by a group of French, American, and British hematologists. These proposals were later enriched by classifications provided by the World Health Organization (WHO), which combined the analyses of morphology, immunophenotyping, cytogenetics, and molecular genetics of acute leukemia cells [44].

### 3.2. Leukemia Subgroups

Based on the type and degree of maturity of the leukemic cell population (blast), which are defined by French-American-British (FAB) classification, AML is further divided into eight subgroups (M0 to M7), including subgroups from granulocytic- or myeloid-derived progenitors (M0 to M3), subgroups from monocytic myeloid-derived progenitors (M4 and M5), and the relatively rare leukemias deriving from megakaryocytic and erythroid progenitors (M6 and M7). The (FAB) classifications provide important guidelines for the diagnosis, treatment, and prognostic prediction of acute leukemia; however, the same classifications fail to provide accurate differentiation between all indicated subtypes, and do not correlate well with the clinical outcomes. The identification of reliable protein biomarkers for acute myeloid leukemia (AML) that could help in diagnosis and prognosis, treatment, and the selection for bone marrow transplant requires substantial comparative proteomic studies and a high number of samples suitable for use in present-day technologies. These efforts have been given new momentum by the more frequent use of high-resolution mass spectrometry combined with newly developed enrichment and labelling schemes to favor the detection and quantification of post-translational modifications. There is no shortage of proteomic studies in the search for protein biomarkers for the diagnosis and classification of AML. That being said, the current literature shows that a major part of these studies has used low-resolution mass spectrometry, low number of investigated samples without follow-up investigations, involving a major number of samples to validate the findings of the initial analyses. However, more recent reports on this argument show that proteomic and phosphoproteomic investigations are taking advantage of more access to large cohorts of AML patients to sample from. It should be pointed out that such proteomic efforts are not limited to biomarkers discovery; a variety of MS-based methods and technologies now play valuable and expanding roles in the diagnosis and monitoring of acute leukemia, as well as in identification of therapeutic targets and biomarkers, drug discovery, and other important areas of leukemia research [45]. 

MS-based approaches to identify and quantify proteins in complex biological samples require a multistep workflow that includes sample preparation, liquid chromatography tandem mass spectrometry (LC-MS/MS), data analysis, and results interpretation. In sample preparation stage, cells are lysed, sulphur bridges reduction, cystine alkylation, protein digestion, and desalting to remove chemicals and residual intact proteins before introducing the sample for LC/MS-MS analysis (see Figure 4).

To perform protein quantification, it is necessary to perform some form of labelling. Stable isotope labeling by amino acids (SILAC) is a metabolic labeling technique that introduces light and heavy versions of amino acids (the most commonly used are arginine and lysine) to whole cells through the growth medium containing the desired amino acid form [46]. Other less used labelling methods include isobaric tags for relative and absolute quantification (iTRAQ) and tandem mass tag (TMT). These two techniques are often used for quantitative biomarker discovery studies where SILAC is not applicable [47].

It is now fairly accepted that AML patients do not exhibit clinical indications, which could help clinicians to predict risk. MS-based proteomics can contribute to the discovery of new prognostic and risk stratifying biomarkers to complement existing morphological, cytogenetic, and molecular risk factors. Several groups have used MS-based proteomics to optimize the cytogenetic classification of AML. An investigation by Nicolas et al. [48] reported the use of protein profiles to subdivide the intermediate and unfavorable cytogenetic groups into subgroups with significantly different survival rates. The same study reported that the most discriminating protein between survival and death was S10A8, which was verified as a biomarker for poor prognosis in a different patient cohort, and found to predict death (during the follow-up period of 57 months) with 85% sensitivity and 72% specificity. Based on these results, the authors suggested that the increased expression of this protein may be used as a high-risk prognostic AML biomarker. In another study, profiling of serum peptides has been performed on samples from 72 AML patients, 72 healthy controls, 37 AML patients with complete remission, and 30 refractory and relapse AML patients [49]. The samples were analyzed with MALDI-TOF and LC-ESI-MS/MS spectrometry to identify candidate biomarkers, followed by immunoblotting for validation studies. Three proteins, ubiquitin-like modifier activating enzyme 1 (UBA1), isoform 1 of fibrinogen alpha chain precursor (FIBA), and platelet factor 4 (PLF4), correlated with AML clinical outcome and could possibly be used for predicting AML relapse, monitoring minimal residual disease, and predicting prognosis in clinical practice. The increase in the availability of AML samples directly from patients or from biobanks, suitable for proteomic analyses, promises new validated protein biomarkers. Furthermore, the increasing use of peptide and modified peptide signatures based on MS data may result in clinical strategies to differentiate AML subgroups and to predict prognosis or treatment response [50].

## 4. Mass Spectrometry in the Study of Solid Tumors in Children

Solid tumors account for about 60% of all pediatric malignant neoplasms. The spectrum of tumor types that occur in children is much different from that observed in adults. Pediatric neoplasms include tumors of the central nervous system (35%), soft sarcoma, including rhabdomyosarcoma (7%), Wilms tumor (6%), bone tumors, including osteosarcoma and Ewing sarcoma (8%), retinoblastoma (5%), and miscellaneous tumors, including hepatoblastoma, germ cell tumors, and melanoma (17%) [51]. Substantial progress has been made in the diagnosis and management of these tumors since the original demonstration of the chemosensitivity of Wilms tumor to actinomycin D in 1966. Cure rates for most childhood solid tumors have increased by as much as 50% since the mid-1970s. Such increase is attributed largely to improved understanding of prognostically important biological and clinical features, enhanced precision of clinical staging systems, consistent use of supportive care, and development of more effective treatment, often incorporating a combination of chemotherapy, surgery, and radiation.

### 4.1. Medulloblastoma

Medulloblastoma (MB) is the most common malignant pediatric brain tumor known to be characterized by high incidence of metastasis of the central nervous system, a characteristic which calls for highly aggressive cure, including prophylactic radiation administered to the entire brain and spine. Unfortunately, such highly aggressive regime leaves the majority of long-term survivors with permanent and debilitating neurocognitive impairments, while the remaining patients relapse with terminal metastatic disease. This dramatic scenario underlines the need for a more targeted and possibly more effective therapies. It goes without saying that to achieve such tantalizing objective, a further understanding of the various mechanisms regulating this disease, particularly the process of metastasis, is necessary. Mass spectrometry-based methods can give a valuable contribution to realize such goal. A recent work by Paine et al. [52] is a representative example in this direction. The authors used MALDI-mass spectrometry imaging to examine whole brains from SmoA1 mice. The authors reported that the disease progression in these species faithfully recapitulates the pattern of tumor progression in humans. Briefly, six mice brains were collected under identical conditions: three brains containing non-metastasizing primary tumor and three brains containing a metastasizing primary tumor. These brains were segmented into three main regions and examined by MALDI-TOF operating in negative ion mode. The mean mass spectra for the three segmented regions from both non-metastasizing and metastasizing MB mouse brains are shown in Figure 5.

The authors identified ten lipids that were differentially expressed within non-metastasizing and metastasizing MB primary tumors. The low abundance and the heterogeneity of these lipids required the imaging of the full 3D volume of the brain and the segmentation of the tumor regions. These lipids were identified as having significant differences in TIC normalized relative abundances between the two groups. Chemical structures of these lipids were obtained using accurate mass measurements and MS/MS experiments. 

Among the lipids observed to increase in metastasizing primary tumors are the phosphatidylinositol and two phosphoinositides. These three lipids are all structurally related, containing the same fatty acyl chain composition and core head group, and differing only by the degree of phosphorylation on the inositol moiety. PIPs are generated by phosphoinositide kinases (PIKs) to convey signals from the cell surface to the cytoplasm and are downstream effectors on multiple pathways involved in the survival and growth of normal cells. Dysregulation of the PI3K signaling pathway, most commonly caused by alteration of the PTEN gene, has been widely associated with oncogenesis in humans due to unbridled cell growth and proliferation. All three lipids—participants in the same signaling pathway– were observed to be elevated in metastasizing tumors compared to their non-metastasizing counterparts, therefore providing strong evidence that this pathway is upregulated in medulloblastoma metastasizing tissue. According to the authors, the ten lipids identified represent prime candidates for validation in human MB tissues and for further investigation with respect to the potential functional mechanism of these lipids in the promotion of metastasis in human and murine SHH MB.

Considering the results reported above, the following observations can be made: (i) Mass spectrometry imaging on its own (without prior separation/fractionation or labelling) can detect specific molecular changes in complex biological samples. (ii) The difference in lipid profiles between metastasizing and none reported in this study is an important contribution to current attempts to understand the mechanism(s) regulating the metastasize and the tumor progression in MB. (iii) Phosphatidylinositol-3 kinases (PI3Ks) constitutes a lipid kinase family characterized by their ability to phosphorylate inositol ring 3′-OH group in inositol phospholipids to generate the second messenger phosphatidylinositol-3,4,5-triphosphate (PIP3) at the inner side of the membrane. It is relevant to point out that PI3K and AKT has gained a relatively recent recognition as an important regulator of mammalian cell proliferation and survival. The dysregulation of several components of this pathway in a wide spectrum of human cancers is one of the reasons why PI3K-AKT has gained a prominent status in cancer research. 

Another recent study demonstrating the use of MS imaging to investigate brain tumors has been reported by Clark et al. [53]. The aim of the study was to identify lipid profiles capable of differentiating medulloblastoma (MB), which originates in the posterior fossa of the brain, from pineoblastoma (PB), which originates within the pineal gland. Medulloblastoma and pineoblastoma exhibit overlapping clinical features and have similar histopathological characteristics. Histopathological similarities confound rapid diagnoses of these two tumor types. The authors used high-resolution Fourier transform ion cyclotron resonance (FT-ICR) equipped with MALDI to examine archived frozen human tumor specimens (eight medulloblastoma and three pineoblastoma tissue samples). The acquired datasets were examined with multivariate statistical analyses to generate classifiers capable of distinguishing the two tumor types. Furthermore, the discriminative molecules were queried against the lipid maps database and identified. The authors concluded that imaging MS identified a number of lipids which could distinguish the two forms of brain cancer. Glycerophosphoglycerols were identified as the classifiers of MB, while sphingolipids were identified as the top classifiers of PB. This study revealed a number of elements which merit further considerations. First, the two investigated forms of cancer are known to exhibit overlapping clinical features and have similar histopathological characteristics, which renders the reported conclusions highly valuable. Second, these conclusions are based on a fairly limited number of samples and therefore, and as suggested by the authors, verification and possible validation of the reported classifiers require further investigations, where much higher number of samples should be used. Third, one of the interesting elements in the same study is that the MS images were serially acquired using alternating positive and negative ion modes. This approach enhances the chances of capturing molecules, which ionizes more efficiently in positive ion mode but not in negative ion mode and vice versa. The authors have used high-resolution MS imaging, which is highly suitable for spatially resolved molecules within a given sample allowing for label free molecular images. This characteristic allows these images to be compared with more traditional histopathology and immunohistochemistry assessment for initial validation.

In a study by Woolman et al. [54], Picosecond infrared laser desorption mass spectrometry (PIRL-MS) was used to generate molecular signatures, which could be used for MB subgroups classification. The authors used in excess of a hundred tissue samples supplied by a local biobank. Each sample was subjected to 10 to 15 s PIRL-MS data collection and principal component analysis with linear discriminant analysis (PCA-LDA). The MB subgroups model was established from 72 independent samples; the remaining 41 samples were classified as unknown tumors and further subjected to multiple 10 s PIRL-MS sampling and real-time PCA-LDA analyses using the above model. The resultant 124 PIRL-MS spectra from each sampling event, after the application of a 95% PCA-LDA prediction probability threshold, yielded a 98.9% correct classification rate. Post-ablation histopathologic analyses suggested that heterogeneity or sample damage prior to PIRL-MS sampling at the site of laser ablation was responsible for the failure of the classification.

### 4.2. Neuroblastoma

Neuroblastoma is the most common extracranial solid tumor in childhood and the most frequently diagnosed neoplasm during infancy. The disease is known for its broad spectrum of clinical behavior and efforts to tailor treatment according to the predicted clinical aggressiveness of the tumor have been ongoing for decades. Neuroblastoma patients are classified according to disease stage and molecular alterations into three groups: low, intermediate, and high risk. Although the first two groups show five-year survival rates greater than 90%, the survival of high-risk patients remains poor at approximately 40%. Despite aggressive treatment consisting of surgery and a combination of high-dose chemotherapy, radiotherapy, and immunotherapy, the survival rate of high-risk neuroblastoma remains notably low [55,56], which renders high-risk NB a good candidate for epigenetic therapies to overcome drug resistance. In fact, epigenetic therapies are an emerging option for overcoming NB drug resistance. This approach proposes targeting of epigenetic regulators, which are proteins involved in the creation, detection, and interpretation of epigenetic signals. Currently, most epigenetic drugs act at three main levels: (i) DNA methylation, which can be modulated by targeting of DNA methyltransferases (DNMT); (ii) histone modifications, such as acetylation and methylation, which can be targeted by inhibiting the enzymes responsible for these chemical changes; and (iii) blockage of the interpretation of these modifications by targeting epigenetic readers, among which proteins containing bromodomains are the most thoroughly characterized.

It is important to point out that there is an unidentified subset of low-risk and intermediate-risk children who would benefit from treatment reduction while maintaining high survival rates. However, current classification fails to distinguish those high-risk NB that will not respond to initial therapy or will relapse after treatment. Thus, a refinement in risk group classification at the time of diagnosis is crucial to guide therapy and identify those patients who will not respond to standard treatments. Increasingly accurate risk classification and refinements in treatment stratification strategies have been achieved with the more recent discovery of robust genomic and molecular biomarkers. The last few years have witnessed increasing use of MS-based strategies in the search for such markers. Two examples, which use such strategies, are considered here. In a study by Quintàs et al. [57], plasma samples collected from 110 NB patients were analyzed by ultra-performance liquid chromatography-time of flight mass spectrometry (UPLC-TOF-MS); the potential of plasma metabolomic profiling to improve current NB risk-group stratification and predict response to therapy was evaluated. Plasma samples were analyzed in positive electrospray ionization mode. A sample aliquot was also analyzed using the auto MS/MS method. The analyses were able to identify different plasma metabolic profiles in high- and low-risk NB patients at diagnosis. According to the authors, the metabolic model correctly classified 16 high-risk and 15 low-risk samples in an external validation set providing 84.2% sensitivity and 93.7% specificity. Metabolomic profiling could also discriminate high-risk patients with active disease from those in remission. Notably, a plasma metabolomic signature at diagnosis identified a subset of high-risk NB patients who progressed during treatment. The main conclusion of this study was the ability of the metabolic profiling to potentially predict those patients with high-risk neuroblastoma who are more likely to progress during treatment.

In a fairly recent study [58], MS imaging was used to generate spatial peptide signatures, which could be used to discriminate NB high from low and intermediate risk. As well as MALDI imaging, the authors also used a bottom-up nano-liquid chromatography electrospray ionization tandem mass spectrometry approach for protein identification. In these analyses, the authors used formalin-fixed, paraffin-embedded (FFPE) tissue sections, in which diagnostic biopsies from primary neuroblastoma categorized them as high (five samples) or other risk groups (low or intermediate risk, four samples). The acquired spatial peptide signatures allowed the identification of 11 proteins, most of which are associated with the extracellular matrix and cytoskeleton, which enabled the authors to distinguish high risk neuroblastomas from the tissue sections independently of conventional histology. Differential expression of the identified discriminative proteins, AHNAK and CRMP1, was immunohistochemically confirmed. These data showed a lower intensity distribution of CRMP1 in high-risk neuroblastomas and inversely higher intensity distribution in low- and intermediate-risk neuroblastomas. The authors claimed that such result is well in line with the reported role of CRMP1 in neuronal differentiation and its previous use as a marker gene in neuroblastoma gene expression panels as well as its usefulness as a prognostic and diagnostic marker in other cancers. The acquired data also identified AHNAK as a marker protein highly expressed in high-risk neuroblastoma, from which tryptic peptides have high intensity distributions in tumor cell-rich regions of sections analyzed by MALDI-MSI. The authors reported that AHNAK has not been previously associated with neuroblastoma but has been implicated in several cellular functions associated with cancer. Furthermore, discriminative spatial intensities of *m*/*z* peaks were validated in microarrayed tissue cores from tumor cell-rich regions in neuroblastomas. The imaging data generated by MALDI-MS allowed the detection of molecular heterogeneity in the investigated regions. In our opinion, the reported detection of molecular heterogeneity in NB by MS imaging needs further investigations, possibly involving much higher number of samples compared to those used in this study. Having said that, these initial data on a limited number of samples are a highly welcomed result. Currently, international risk classification of neuroblastoma is mainly based on clinical criteria plus MYCN oncogene amplification and more recently complemented by transcriptomic data, proving its central role for making therapy decisions and for disease management. Adding diagnostic information derived from proteomic analyses can only improve the precision of current risk classification approaches. The data presented by Wu et al. [58] provide a proof-of-concept for the technical feasibility of this approach. Furthermore, detection of tumor heterogeneity reported in the same study is likely to be a useful contribution to future efforts aimed at more reliable selection of prognostic or predicative biomarkers and signatures at the protein level in neuroblastoma.

Recent progress in epigenetic research has given hope for cancer patients, particularly those suffering from forms resistant to existing conventional therapies. Epigenetic therapies are generally designed to reverse the oncogenic alterations in chromatin components. These therapies are considered an emerging alternative to existing therapies against aggressive tumors that are or will become resistant to conventional treatments. In principle, these therapies target regulators, which are proteins that are involved in the creation, detection, and interpretation of epigenetic signals, such as methylation or histone post-translational modifications. MS-based proteomics is giving a substantial contribution to current efforts to investigate and understand these protein modifications. A recent article [59] described MS-based workflow that is capable of tracking all possible histone PTMs in an untargeted approach that makes use of human cells. In this workflow, histones were extracted from harvested cell culture, chemical derivatization by propionylation, and tryptic digestion followed by LC/MS, MS-MS. The derivatization step was necessary [60] because of the unusually high levels of lysines and arginines within the sequence of histone proteins; such high levels have the consequence that proteins digestion by trypsin would result in relatively short tryptic peptides and would vary based on the modification status of the histones. Derivatization by propionylation has two benefits: Firstly, it modifies all free primary amines, resulting in an Arg-C specificity instead of a tryptic specificity. Secondly, short tryptic peptides will show less retention on a C18 column during reversed-phase high-performance liquid chromatography. The presence of a propionyl group increases the hydrophobicity, thereby enhancing the retention and separation of the peptides in the LC analyses. That being said, the optimization of the derivatization step is not as straight forward as it sounds. In a careful examination of the various derivatization procedures [59], the authors investigated the pitfalls in histone propionylation during bottom-up mass spectrometry analysis. 

## 5. Epigenetic-Based Therapies

Epigenetic therapies are an emerging alternative for overcoming drug resistance. This approach proposes targeting of epigenetic regulators, which are proteins involved in the creation, detection, and interpretation of epigenetic signals. The reversible nature of the epigenetic changes that occur in cancer has led to the possibility of ‘epigenetic therapy’ as an alternative to conventional therapies. The aim of epigenetic therapy is to reverse the epigenetic aberrations that occur in cancer, leading to the restoration of a ‘normal epigenome’. A number of epigenetic drugs have been discovered in the last ten years that can effectively reverse DNA methylation and histone modification aberrations that occur in cancer. DNA methylation inhibitors were among the first epigenetic drugs proposed for use as cancer therapeutics. The potential of these drugs was realized over 50 years ago with the publication by Constantinides [61], showing that treatment with cytotoxic agents, 5-azacytidine (5-aza-CR) and 5-aza-2′-deoxycytidine (5-aza-CdR), leads to the inhibition of DNA methylation that induced gene expression and caused differentiation in cultured cells. The authors are not in a position to list and discuss the epigenetic mechanisms, which led to the discovery and development of various drugs, which gave hope to cancer patients, in particular those in pediatric age. That being said, the following general observations are in order: The recent literature has amply demonstrated that these epigenetic modifications are central in the initiation and progression of various forms of cancer. Furthermore, the information on the type and on the position of these modifications provided by MS-based proteomics are highly relevant to current and future efforts to discover new epigenetic therapies. However, further analytical data are needed before considering such technology a source for diagnostic and prognostic biomarkers to suggest therapeutic path for various forms of cancer. Despite the great potential of DNA methylation inhibitors as therapeutic option, their reaction with the DNA is known to be non-specific, causing genome-wide hypermethylation, which results in undesirable effects regarding the activation and/or silencing of various genes. It has been argued that such side effect could be mitigated by chemically synthesized small molecules, which are more effective than cytidine analogues. The attractive characteristic of these small molecules is that they do not require incorporation into DNA and bind directly to the catalytic site of the DNA methyl transferases [62].

In recent years, it has been established that aberrant gene silencing is linked to histone acetylation, bringing histone acetylation pattern to its normal state through treatment with histone deacetylase inhibitors (HDAC). These agents have been shown to have antitumor effects, including growth arrest, apoptosis, and the induction of differentiation. These effects by HDAC inhibitors are due to their ability to reactivate silenced suppressor genes [63]. A representative member of this class of inhibitors is suberoylanilide hydroxamic acid (SAHA), which has been approved for use in clinic over fifteen years ago for the treatment of T cell cutaneous lymphoma [64]. The well-studied interaction between the different components of the epigenetic machinery renders the use of combinatorial cancer strategies a highly attractive therapeutic option. A representative example of this combinatorial cancer treatment is the use of both DNA methylation in combination with HDCA inhibitors. Synergistic activities of DNA methylation and HDAC inhibitors were also demonstrated in a study showing greater reduction of lung tumor formation in mice when treated with phenylbutyrate and 5-Aza-CdR together [65].

## 6. Mass Spectrometry Monitoring of Therapeutic Drugs

Chemotherapeutic drugs are still a principal player in the treatment of various forms of cancer, particularly those in advanced stages. The determination of appropriate dosing regimens for the treatment of infants and very young children with cancer represents a major challenge in pediatric oncology. Dose reductions are commonly employed for many chemotherapeutic regimes; however, the suitability of dose reductions for various anticancer drugs remains unclear due to the lack of deep knowledge of the pharmacokinetics of these drugs in the infant patient population. In any disease, the efficacy of a given drug is measured by considering both its therapeutic effect and its toxicity. It is also well established that most anticancer drugs are cytotoxic in nature and have low specificity, which means that the optimal range of a therapeutic dose for a given patient is likely to be narrow. Recent works on this argument have shown that for most anticancer drugs, the concentration at which serious toxicity tends to manifest is many times higher than the concentration at which the therapeutic efficacy is achieved. This observation underlines the central role of therapeutic drug monitoring (TDM). Numerous works have demonstrated that failure of systemic cancer treatment can be, at least in part, due to the drug not being delivered to the tumor at sufficiently high concentration and/or sufficiently homogeneous distribution. The last decade has witnessed an increased use of mass spectrometry for TDM analyses. Aghai et al. [66] used LC/MS-MS for the simultaneous determination of ten kinase inhibitors in human serum and plasma. This method was designed for application in daily clinical routine. The authors reported that the development and validation of the method was according to the US Food and Drug Administration (FDA) and European Medicines Agency (EMA) validation guidelines for bioanalytical methods. The main steps in this method included proteins precipitation of the plasma samples and room temperature LC separation; eluted components were injected into positive electrospray ion source and the resulting ions were analyzed and detected using multiple reaction monitoring mode. Stable isotopically labeled compounds of each kinase inhibitor were used as internal standards. This method has a number of interesting elements, which should be underlined: (i) The method has a relatively short analysis time (7 min), which renders it highly suitable for high throughput analyses, particularly in clinical environment. (ii) A number of therapeutic regimes use multiple anticancer drugs simultaneously, which renders the simultaneous monitoring of multiple compounds highly informative regarding the response of the patients and the therapeutic usefulness of the combination of these drugs.

The relatively recent introduction of oral anticancer drugs resulted in significant improvement in the treatment of various forms of cancer. Patients undergoing this type of therapy tend to have variable concentrations of plasma, resulting in reduced therapeutic efficacy of the drug, and in some cases, unforeseen side effects One approach to mitigate these effects is regular TDM analyses of the patients receiving such treatment. Kehl et al. [67] described a method to simultaneously quantify the plasma concentrations of 57 oral antitumor agents. The authors used liquid chromatography coupled to high-resolution mass spectrometry and the method was fully validated according to the FDA guidelines. The application of this method to clinical routine was tested by the analysis of 71 plasma samples taken from 39 patients.

Mass spectrometry imaging (MSI) is a label-free molecular imaging technique that provides spatial as well as temporal information on the spatial distribution of drugs and their metabolites in a wide range of biological samples, a capability which lacks in the more diffused LC/MS methods. In recent years, (MSI) started to give relevant contribution to the emerging field of precision pharmacology. In this approach, various analytical techniques, including MSI, are used to understand whether an administered anticancer drug is being adequately delivered to the tumor region. Such information is central to both the assessment of the adequacy of the therapy and the resistance to the drug. In a relatively recent study [68], the authors reviewed the role of MSI in pre-clinical studies to characterize anticancer drug distribution within the body and the tumor, and the application of MSI in pre-clinical studies to define optimal drug dose or schedule, combinations, or new drug delivery systems. In their review, the authors underlined the following observations: the characterization of drug concentrations, and in particular drug distribution, within tumors or normal tissues is a challenge that the rapidly developing technique of MALDI-MSI has started to address and may have an important role in drug development. MALDI-MSI already has the potential to support pre-clinical studies by comparing the penetration of candidate molecules into different regions of spheroids in vitro and xenografts in vivo.

Published works in the last few years clearly show that LC coupled to tandem mass spectrometry is the preferred method for TDM analyses. It is also interesting to note that these works indicate a limited use of high-resolution mass spectrometry. This limited use can be justified by the following considerations: First, TDM analyses detect and quantify chemical entities with known molecular structure(s) and predetermined molecular weights; therefore, instruments with mass resolution of one atomic mass unit (amu) are sufficient for this type of analyses. This explains the extensive use of triple quadrupole instruments hyphenated with liquid chromatography. That being said, certain analyses conditions do necessitate the use of high-resolution mass spectrometry. For example, simultaneous monitoring of multiple anticancer drugs, some of which happen to have the same nominal molecular mass but different elemental composition. Identification of some untargeted metabolites of the monitored compounds is another analyses condition in which high resolution can be the only route for an unambiguous identification of the detected metabolites. Second, the relatively limited use of high-resolution instruments for TDM analyses can be partially attributed to higher costs and more demanding operational skills compared to their low-resolution counterparts.

## 7. Conclusions and Perspectives

Works discussed in this review underline the central role of MS-based proteomics in cancer research. The cited examples show how such approach is providing a wealth of information relevant to various aspects of cancer research, including biomarkers discovery, in particular those which could be used to discriminate tumor subgroups which happen to have similar or even overlapping histopathological features. The same approach furnishes much needed information on metabolic profiles, therapeutic drug monitoring, protein post-translational modifications, and molecular mechanisms in cancer, and signaling pathways associated with various forms of cancer. In recent years, such contribution has been enhanced by more frequent use of high-resolution mass spectrometry in combination with more efficient separation, fractionation, and metabolic labelling methods. MS-based proteomics has been given a fresh momentum by recent epigenetic findings, showing the central role of protein modifications in the landscape of epigenetic events. The involvement of epigenetic abnormalities in the initiation and progression of cancer is now widely accepted. Consequently, targeting the enzymatic machinery that controls the epigenetic regulation of the genome has emerged as a highly promising strategy for therapeutic intervention, in particular for forms of cancer showing drug resistance. It is also becoming more evident that the development of epigenetic drugs requires a detailed knowledge of the processes that govern chromatin regulation. Mass spectrometry-based proteomics is becoming a major source for such knowledge. There are various scientific opinions predicting that in the next few years more cancer therapies will be approved, as many are currently in advanced clinical trials. The authors are convinced that MS-based proteomics will greatly contribute to such anticipated developments.

## Figures and Tables

**Figure 1 medicina-59-00612-f001:**
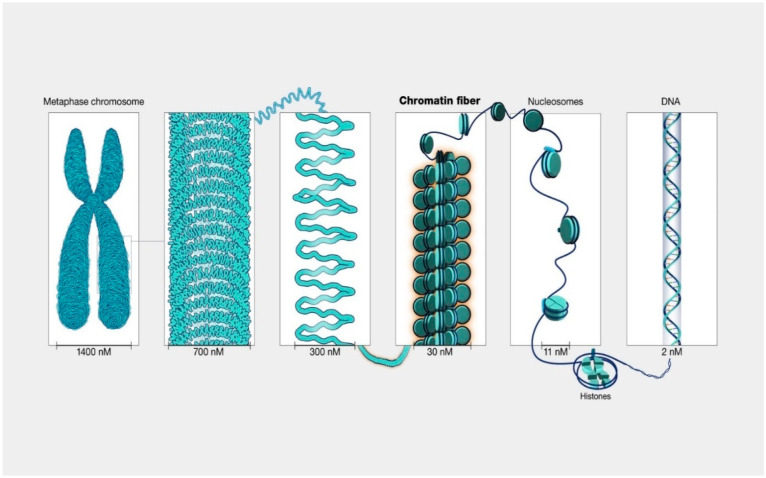
The structure of chromatin within a chromosome. Chromatin refers to a mixture of DNA and proteins that form the chromosomes found in the cells of humans and other higher organisms. Many of the proteins, namely histones, package the massive amount of DNA in a genome into a highly compact form that can fit in the cell nucleus. Courtesy: National Human Genome Research Institute.

**Figure 2 medicina-59-00612-f002:**
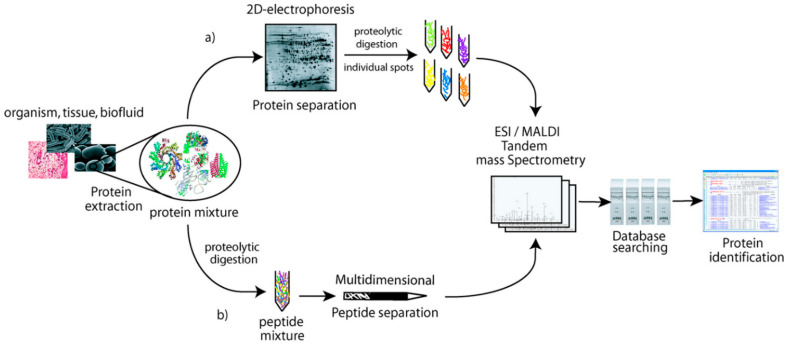
The main steps in bottom-up analyses of a protein mixture, Two routes of analysis: the upper shows digestion of individual proteins separated by 2D-PAGE, while the lower route shows the digestion and analysis of protein mixture. (Reprinted with permission from Fournier ML, Gilmore JM, Martin-Brown SA, Washburn MP. Multidimensional separations-based shotgun proteomics. *Chem. Rev.*
**2007**, *107*, 3654–3686. Copyright {2007} American Chemical Society).

**Figure 3 medicina-59-00612-f003:**
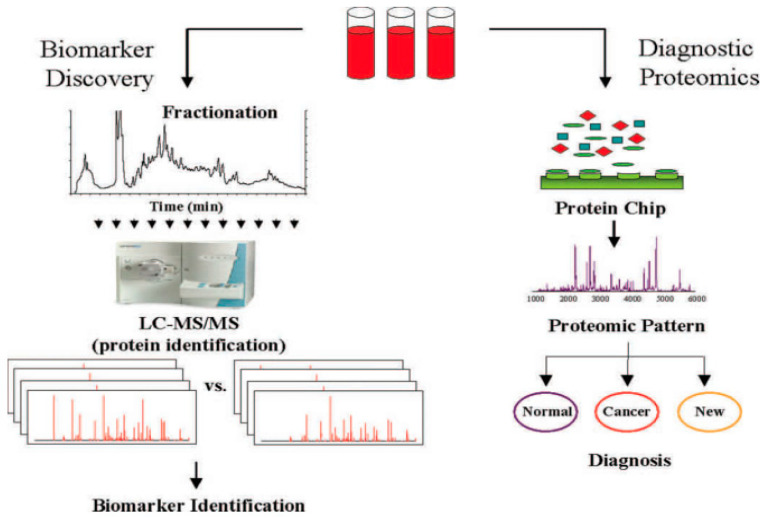
MS-based methods for biomarker discovery (left side) and pattern-based diagnostic proteomics (right side). (Reprinted/adapted with permission from Ref [38]. Copyright year 2005).

**Figure 4 medicina-59-00612-f004:**
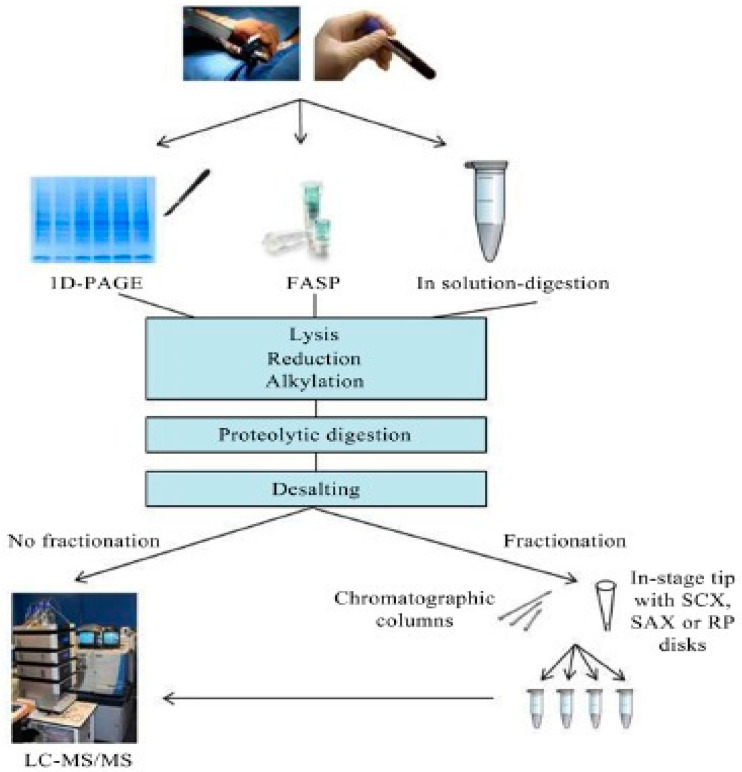
Proteomic workflow comprising three approaches for sample preparation, including filter-assisted sample preparation (FASP). (From Aasebø, E.; Forthun, R.B.; Berven, F.; Selheim, F.; Hernandez-Valladares, M_._ Global Cell Proteome Profiling, Phospho-signaling and Quantitative Proteomics for Identification of New Biomarkers in Acute Myeloid Leukemia Patients. *Curr. Pharm. Biotechnol*. **2016**, *17*, 52–70).

**Figure 5 medicina-59-00612-f005:**
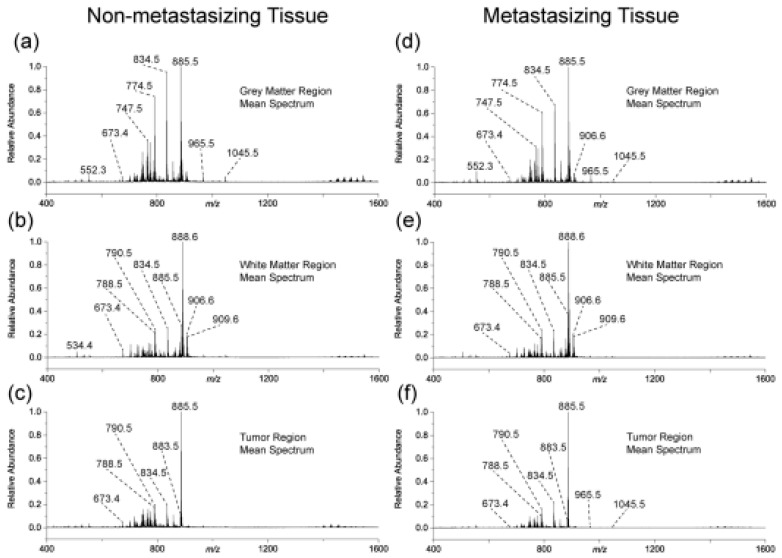
The calculated mean spectra for tumor regions mouse brain containing a non-metastasizing medulloblastoma primary tumor (**a**–**c**) and tumor regions in a mouse brain containing a metastasizing medulloblastoma primary tumors (**d**–**f**). (From Paine et al., *Sci. Rep.* 2019).

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
