# Peer review of "Mass Spectrometry Contribution to Pediatric Cancers Research"

_medicina, 2023, doi:10.3390/medicina59030612_

Round 1

Reviewer 1 Report

The manuscript entitled "Mass Spectrometry contribution to pediatric cancers Research" by Marco Agostini et al have reviewed on recent contribution of mass Spectrometry  on pediatric  cancers. 

The manuscript were very written very well and recent literature were described. 

Some typo errors are found in the mansucript

In Page 1 Introduction section line 35 no 

In page 13 line no 545-561  format the sentence.

Additionally, the author could have included more information on how mass spectrometry-based proteomics works and how it is used in cancer research. Overall, the manuscript provides a good overview of the topic, but there is room for improvement in the writing style and depth of explanation.

Author Response

We wish to thank for the valid and useful suggestions.

1)The typo errors have been amended;

2) in page 13 the sentence has been formatted.

Reviewer 2 Report

In this manuscript, the authors provide a review of current applications of mass spectrometry in pediatric cancer research. Strengths of the manuscript include detailed technical explanations, many specific examples, and a good historical perspective. The manuscript would benefit from a more specific statement of purpose, adjustments in organization, and minor improvements such as clarifying abbreviations. 

General comments:

1) Please provide a more precise statement of what the review is intended to cover, ideally in the introduction. Example: "This review will discuss the history and current applications of mass spectrometry in pediatric cancer research, with particular emphasis on epigenetics in tumor biology and therapy."

2) Overall, the level of detail is good, but the details given in some areas are superfluous. These are described further in the specific comments.

3) As the authors point out, sarcomas are common pediatric solid tumors. Does MS have a role in sarcoma research or diagnostics?

Specific comments:

1) Line 107: what does the abbreviation PTM stand for?

2) Line 111: I assume LC stand for liquid chromatography, but please specify

3) The paragraph on cisplatin therapy and mitochondrial protein detection by 2D-DIGE and MALDI-TOF-MS (line 146) can be shortened substantially. Focus on the role of mitochondria in cisplatin resistance and the role of MS in detecting relevant proteins.

4) Section 3.2, leukemia subgroups, paragraphs 2 (lines 500-505) and 3 (lines 512-518). Are the technical details described in these paragraphs relevant to leukemia specifically? These paragraphs seem slightly out of place.

5) Section 4, line 545. This paragraph is a different font, and appears hyperlinked. Please fix this.

6) Epigenetic-based therapies (line 775) should be section 5, not section 4.

7) Mass spectrometry monitoring of therapeutic drugs (line 820) should be section 6, not section 4.

Author Response

The authors thank sincerely for the valid suggestions, following which the manuscript has been amended:

1) a more precise statement of the aim of the review has been inserted;

2) PTM and LC abbreviations have been specified;

3) The paragraph on cisplatin therapy and mitocondrial protein detection has been left, considering that it was a good example of what can be obtained by the cited MS methods;

4) The technical approaches employed are of general use, but they worked properly in the case of leukemia subgroups;

5) the font of section 4 has been substituted with the right one;

6) the numbering of sections 5 and 4 has been changed with the right ones (5 and 6)

Reviewer 3 Report

In this manuscript Traldi and co-workers present a concise summary of the role of mass spectrometry in pediatric cancers research. Several recent investigations reporting the key mechanisms behind the initiation and progression of various forms of cancer have been discussed. This review is a nice summary and useful addition to literature however some revisions are required before publication.

-Proper capitalization of words in title is required.

-The figures should be provided with better resolution.

-The reference numbering should not be superscript. Follow guidelines.

-line 545-561, check font of text here and at several other places like in figure captions etc.

-Several technical sections at the end of manuscript are missing.

-The presentation of references is completely out of format. Some are underlined. Journals are abbreviated while others left with their full names and many more little errors. Follow guidelines.

-

Author Response

I wish to thank for the useful suggestions given:

1) proper capitalization of words in the title has been done;

2) The references numbers have been inserted in brackets;

3) The font has been checked in all the manuscript;

4) the presentation of the references has been done following the guidelines